# Highlighting the Major Role of Cyclin C in Cyclin-Dependent Kinase 8 Activity through Molecular Dynamics Simulations

**DOI:** 10.3390/ijms25105411

**Published:** 2024-05-15

**Authors:** Sonia Ziada, Julien Diharce, Dylan Serillon, Pascal Bonnet, Samia Aci-Sèche

**Affiliations:** 1Institut de Chimie Organique et Analytique (ICOA), UMR CNRS-Université d’Orléans 7311, Université d’Orléans BP 6759, 45067 Orléans CEDEX 2, Francepascal.bonnet@univ-orleans.fr (P.B.); 2Université Paris Cité and Université des Antilles and Université de la Réunion, INSERM, Biologie Intégrée du Globule Rouge, UMR_S 1134, DSIMB Bioinformatics Team, 75014 Paris, France; julien.diharce@u-paris.fr

**Keywords:** CDK8 cyclin C, protein–protein interaction, molecular dynamic simulation, free energy calculation, drug design

## Abstract

Dysregulation of cyclin-dependent kinase 8 (CDK8) activity has been associated with many diseases, including colorectal and breast cancer. As usual in the CDK family, the activity of CDK8 is controlled by a regulatory protein called cyclin C (CycC). But, while human CDK family members are generally activated in two steps, that is, the binding of the cyclin to CDK and the phosphorylation of a residue in the CDK activation loop, CDK8 does not require the phosphorylation step to be active. Another peculiarity of CDK8 is its ability to be associated with CycC while adopting an inactive form. These specificities raise the question of the role of CycC in the complex CDK8–CycC, which appears to be more complex than the other members of the CDK family. Through molecular dynamics (MD) simulations and binding free energy calculations, we investigated the effect of CycC on the structure and dynamics of CDK8. In a second step, we particularly focused our investigation on the structural and molecular basis of the protein–protein interaction between the two partners by finely analyzing the energetic contribution of residues and simulating the transition between the active and the inactive form. We found that CycC has a stabilizing effect on CDK8, and we identified specific interaction hotspots within its interaction surface compared to other human CDK/Cyc pairs. Targeting these specific interaction hotspots could be a promising approach in terms of specificity to effectively disrupt the interaction between CDK8. The simulation of the conformational transition from the inactive to the active form of CDK8 suggests that the residue Glu99 of CycC is involved in the orientation of three conserved arginines of CDK8. Thus, this residue may assume the role of the missing phosphorylation step in the activation mechanism of CDK8. In a more general view, these results point to the importance of keeping the CycC in computational studies when studying the human CDK8 protein in both the active and the inactive form.

## 1. Introduction

Cyclin-dependent kinases (CDKs) are serine-threonine kinases that require binding with regulatory proteins called cyclins to be active. CDKs are the main regulators of the cell cycle and gene transcription. The human proteome contains 20 CDKs and 29 cyclins. CDK1 to CDK6 are involved in cell cycle regulation, while CDK7, CDK8, CDK9, CDK11, and CDK20 are primarily involved in transcriptional regulation. More particularly, CDK7, CDK8, and CDK9 control the activity of RNA polymerase II in humans through the phosphorylation of its C-terminus domain, which catalyzes the synthesis of all mRNA precursors [1]. The inhibition of CDK activity by small molecules for the treatment of cancer has been extensively studied [2]. Several CDK inhibitors have undergone clinical trials, and, in February 2015, palbociclib, a CDK4/6 inhibitor, was first approved by the FDA [3].

CDK8 is a target of interest that has recently attracted considerable attention after the publication of numerous genetic and biochemical studies highlighting its many key roles in oncogenesis [4,5]. Among its various cellular functions, the most notable is its involvement in regulating transcription through diverse mechanisms. CDK8 is a part of the mediator complex, which is a large, multi-subunit protein complex that is central to the regulation of transcription in eukaryotes [6]. The main function of the mediator complex is to transmit regulatory signals from DNA-bound transcription factors to the RNA polymerase II (RNAPII). The complex CDK8–cyclin C (CDK8–CycC) associates with mediator complex subunits 12 (MED12) and 13 (MED13) to form the CDK8 module, a sub-module of the mediator complex [7,8,9]. In humans, it has been demonstrated in vitro that the CDK8 module inhibits the initiation of transcription by deactivating CDK7, which can no longer phosphorylate the carboxy-terminal domain of RNAPII, thereby blocking the transcription [10]. On the other hand, contrary to this transcriptional repression role, a positive regulatory role for CDK8 via the recruitment of the SEC (Super Elongation Complex) has been observed in vivo. In fact, the interaction of the mediator complex with SEC facilitates the elongation and release of certain genes [11,12]. In particular, CDK8-mediated activation of the Wnt-β–catenin signaling pathway [13] and the transcription of estrogen-inducible genes [14] contribute, respectively, to oncogenesis in colorectal and mammary tumors, making CDK8 an oncogene of interest.

Since Schneider et al. published in 2011 the first crystallographic structure of human CDK8–CycC complexed with sorafenib (PDB ID: 3RGF) [15], a total of 31 experimental structures are currently available. All of these crystal structures present 10 to 20 missing residues within a region that lies outside of the active-site cleft called the activation loop. This motif has a central role in regulating the activity of protein kinase by generally adopting a *DFG-in* conformation in the active form and a *DFG-out* conformation in the inactive form [16], with DFG referring to the Asp-Phe-Gly sequence at the beginning of the activation loop. In that connection, the first computational study on human CDK8 (with PDB ID: 3RGF) aimed at providing insights into two point mutations within the activation loop through 50 ns of all-atom conventional molecular dynamics (cMD) simulation in implicit solvent [17]. Moreover, the theoretical binding free energy between CDK8 and CycC was also determined using the molecular mechanics Poisson–Boltmzann surface area (MM-PBSA) and the molecular mechanics generalized Born surface area (MM-GBSA) methods on the basis of 2 ns of all-atom cMD simulation in explicit solvent. However, in in silico structural studies, particular attention should be paid to the building of a relevant model of the protein, especially in a study [17] where the object of the investigation, the activation loop, is missing and has to be reconstructed. Surprisingly, the authors used a template where the activation loop is in the *DFG-in* conformation to model the activation loop of 3RGF (PDB ID), which is in the *DFG-out* conformation. Cholko et al. studied twelve CDK8–CycC systems using 500 ns all-atom cMD simulations in explicit solvent with the aim of elucidating the system motions and the structural determinants that affect protein–ligand interactions [18]. They found that the CycC is important in providing proper interactions for ligand binding, whereas the highly flexible activation loop has little effect. Furthermore, they employed MM-PBSA analysis to characterize protein–ligand interactions from an energetical point of view and discussed the major driving force of protein–ligand binding.

In this study, we investigated the effect of CycC on the structure and dynamics of CDK8, on the one hand, and, on the other hand, the structural molecular basis of the protein–protein interaction between the two partners. Indeed, the presence of CycC in the CDK8–CycC complex seems to play a more complex role than for other members of the CDK family [19,20]. CDKs are generally activated in two steps: (1) the binding of the cyclin (Cyc) to CDK, and (2) the phosphorylation of a threonine residue in the CDK activation loop (T160 in human CDK2). The binding of the Cyc to CDK induces a conformational change in the αC-helix, which adopts an *αC-helix in* conformation (shift toward the binding site) from an *αC-helix out* conformation. The phosphorylated threonine on the activation loop serves as an anchor for adjusting the orientation of three conserved arginine residues, inducing a conformational change in the activation loop that shifts from a *DFG-out* to a *DFG-in* conformation [21]. In CDK8, the phosphorylation step has not been observed and is not required for its activation [22,23]. Moreover, the first published crystallographic structures of human CDK8–CycC [15,24] and a more recent one [25] display a surprising conformation corresponding, somehow, to the “intermediate state of the activation mechanism”. Indeed, the αC-helix is in *αC-helix in* conformation, which is expected as CycC is bound to CDK8 in agreement with the activation mechanism. However, the phosphorylation step did not occur due to the lack of the conserved threonine in the CDK8 sequence [22], leading to keeping a *DMG-out* conformation (in CDK8, a Asp-Met-Gly (DMG) motif replaces the well-known DFG motif of protein kinases) for the activation loop. All of these structures are co-crystallized with an inhibitor, which is said to be responsible for the conformational change from the *DMG-in* to the *DMG-out* conformation. Protein kinase inhibitors are classified based on their binding to their receptor [26]. Type I inhibitors bind to the ATP binding site, and type II inhibitors extend from the ATP binding site into a neighboring pocket, the allosteric pocket (also called the “hydrophobic pocket”), which is only accessible through the rearrangement of the DFG motif from the *DFG-in* to the *DFG-out* conformation. The type III inhibitors bind only to the allosteric pocket. All co-crystallized inhibitors of CDK8 belong to the type II or type III class of protein kinase inhibitors. As far as we know, CDK8 is the only CDK family member for which the following structure is obtained experimentally: a *DFG-out* conformation (*DMG-out* in CDK8) while being associated with CycC. All CDK structures complexed with Cyc are usually in *DFG-in* conformation in accordance with its activation mechanism. Alexander et al. tried to reproduce this particular conformation with the complex CDK2–CycB. They incubated the CDK2–CycB complex with a type II inhibitor and also observed a *DFG-out* conformation. However, they found that binding of a type II inhibitor to CDK2–CycB results in the dissociation of cyclin B from CDK2 in a competitive manner [27]. All of those observations raise the question of the role of CycC in the complex CDK8–CycC in the inactive conformation (*DMG-out*). In particular, it is interesting to investigate whether the CycC has an impact on the structure and dynamics of CDK8. In addition, this impact is the same in active (*DMG-in*) and inactive (*DMG-out*) conformations and in the presence and absence of the ligand, which has to be explored. Furthermore, in view of this unique capability of CDK8 to bind the CycC in both conformations, it is relevant to study the interaction between CDK8 and CycC in order to decipher the interaction on a molecular basis and to highlight possible important CDK8-specific interaction hotspots. We have noticed that the particular behavior of the CycC among the cyclin family has already been raised before. This led Barette et al. [28] to manage mutagenesis experiments that highlighted a double point mutation of R65A/E66A in CDK8 that greatly affects its capacity to bind to CycC. This effect was partly explained by the X-ray structure, which shows the contacts between Met61 and Arg65 in the human CDK8 and CycC.

Through molecular dynamics (MD) simulations and binding free energy calculations, we found that CycC has a stabilizing effect on CDK8, and we also noted the importance of CDK8 for maintaining a proper conformation in the active and inactive form of CDK8–CycC. The per residue free energy decomposition method enabled us to characterize the CDK8–CycC binding surface, identify the important residues, and obtain their energy contributions. We found that CDK8–CycC presents specific interaction hotspots within its interaction surface compared to other human CDK/Cyc pairs. Targeting these specific interaction hotspots could be a promising approach in terms of specificity to effectively disrupt the interaction between CDK8 and CycC and thus to interfere with the function of CDK8 as an oncogene. The simulation of the conformational transition from the inactive to the active form of CDK8–CycC through targeted molecular dynamics (TMD) simulation suggests another mechanism that could substitute the missing phosphorylation step in the activation mechanism of CDK8. In a more general view, these results point to the importance of keeping the CycC in computational studies when studying the human CDK8 protein in both the active and the inactive forms.

## 2. Results and Discussion

To investigate the effect of the CycC on the CDK8’s behavior, eleven systems were simulated by considering different conformations of CDK8 (*DMG-out*, *DMG-in*) in the presence or absence of CycC, in the presence or absence of the ligand, and whether it is WT or mutated. A description of these systems is provided in the Section 3.

### 2.1. Effect of CycC Exclusion on Structure and Dynamics of CDK8

In order to evaluate the effect of CycC on the structure and dynamics of CDK8, the trajectories were analyzed in pairs (with/without Cyc, that is, 1a/2a, 1b/2b, 3/4, 6/7, and 8/9), as shown in Table 1. For the systems in the *DMG-out* conformation, the average root mean square deviation (RMSD) is higher in the absence of the CycC, which means that the CDK8 structure deviated more from its crystallographic structure in the absence of the CycC. For the system in the *DMG-in* conformation, the average RMSDs are comparable with and without CycC.

The root mean square fluctuation (RMSF) plots (Figure 1 and Appendix A) indicate that the absence of CycC increases the motions of one or more of these regions of CDK8: (1) the αC-helix in all cases, which is in direct interaction with the CycC, (2) the αB-helix in all cases, except in the system 1b (the αB-helix is also in direct interaction with the CycC), and (3) the activation loop in all cases, except in the system in the *DMG-in* conformation (system 9).

In the absence of CycC, the αC-helix has a larger degree of motion, and it can move toward the region normally occupied by the CycC to adopt an *αC-helix out-like* conformation (Appendix A). The αB-helix tends to bend toward CDK8 and interact with it; in system 1b, this leads to its stabilization (Appendix A).

In order to provide a global view of the effect of the presence of CycC on the structure of CDK8, each pair of trajectory systems with and without CycC (1a/2a, 1b/2b, 3/4, 6/7, and 8/9) was combined in a single trajectory by extracting the backbone coordinates of CDK8 from both trajectories. Then, a principal component analysis (PCA) was applied using the conditions described in the Materials and Methods section to each combined trajectory to see if the conformations coming from the simulation with CycC differ from those coming from the simulation without CycC.

In all five cases, we observe two groups formed along PC1 that correspond to the CDK8 conformations extracted from the simulations in the presence and absence of CycC. An example of the PCA projection is presented Figure 2 for two combined trajectories among the five. The PC1 (principal component 1) is thus able to separate the CDK8 conformations according to the presence or not of the CycC in the simulation. We also notice a larger scattering of the CDK8 conformations obtained in the absence of CycC compared to the ones generated in presence of CycC. This indicates an increase of CDK8 conformational sampling in the absence of CycC. Moreover, in all cases, the two first PCs capture more than 60% of the variance, and PC1 alone represents more than 50% of the variance. Considering these results together, it appears that PC1 has captured the regions of CDK8 in which the structure is the most affected by the presence/absence of CycC. It is therefore interesting to analyze the contribution of each residue of CDK8 to PC1, commonly called the “loading plot”, to identify these regions (Figure 3).

First of all, we remark that the PCA loading curves of *DMG-out* conformation systems display a good alignment (Figure 3, bottom left). Second, a common point emerges from all PCA loading curves: the αB-helix and the activation loop contribute greatly in both cases (*DMG-in* and *DMG-out* conformation systems) to separating the structures coming from the simulations performed with and without CycC. This means that the activation loop and the αB-helix adopt different conformations depending on whether CycC has been kept or not. Note that both the activation loop and the αB-helix may adopt several conformations even in the absence of CycC. As seen previously through the analysis of the RMSF, the activation loop and the αB-helix are more flexible in the absence of CycC. On average, their conformations sampled in the absence of CycC are significantly different from those adopted in the presence of CycC. Other regions contribute at varying levels in the *DMG-in* and *DMG-out* conformation systems to separating the two groups (with and without CycC), such as the αF-αG loop, which contributes greatly in the *DMG-in* conformation system but not in the *DMG-out* conformation systems and vice versa for the αD-αE loop. It is also interesting to note that the αC-helix, which is more flexible in the absence of CycC (cf. Figure 1 and Appendix A), does not show a significantly different conformation in the absence of CycC for the *DMG-out* conformation systems (Figure 3).

In conclusion of this part, RMSF plots show that CycC stabilizes the αC-helix in both *DMG-in* and *DMG-out* conformation systems and the activation loop of CDK8 in the *DMG-out* conformation system. It also reduces the fluctuations of the αB-helix, but, in some cases, no difference was observed between systems with/without CycC because the αB-helix bends toward CDK8 and stabilizes itself (Appendix A). The PCA analysis was able to separate CDK8 structures coming from the simulation performed with and without CycC, which highlights an effect of the CycC on the conformation of CDK8. In particular, the CycC greatly affects the conformation adopted by the αB-helix and the activation loop. CycC also impacts the dynamics of CDK8 as the greatest amplitude motions within CDK8 are not the same depending on whether CycC is present or not (Appendix A). In the literature, Cholko et al. [18] also pointed out the importance of the CycC for maintaining the proper structure and dynamics of CDK8. Through MD simulation on 12 CDK8–CycC systems (6 of *DMG-in* conformation and 6 of *DMG-out* conformation), they observed that CycC stabilizes CDK8 by reducing the fluctuations of the αB-helix, the αC-helix, and the activation loop. They also mentioned that the αC-helix adopts an *αC-helix out* conformation in the absence of the CycC, and they pointed to the importance of the CycC for maintaining proper protein–ligand interaction. Concerning this last point, we also find that the CycC stabilizes the ligand in the binding site (Appendix A). In a more general view, these results highlight the importance of keeping the CycC in computational studies.

### 2.2. Understanding the Molecular Basis of the Interaction between CDK8 and CycC

#### 2.2.1. CDK8–CycC Binding Free Energy

To compute the binding free energy of CycC to CDK8 and gain insights into the binding interaction surface, the MM-GBSA approach was applied on the 9500 snapshots extracted from the trajectories in the range of 50 ns–1 μs (i.e., one snapshot every 100 ps). We want to know whether CycC has a stabilizing effect in terms of binding free energy in (1) the active form of the CDK8–CycC complex (with CDK8 in the *DMG-in* conformation), (2) the inactive form of the complex (with CDK8 in the *DMG-out* conformation), and (3) the mutated form of the complex CDK8^R65A-E66A–^CycC. In the presence of Cyc, the active form of the CDK–Cyc complex is the form commonly observed in the crystallographic structures of human CDK family members, in agreement with the general activation mechanism of CDKs. In contrast, the inactive form of the CDK8–CycC complex is the first experimental structure exhibiting such a conformation. The mutant CDK8^R65A-E66A–^CycC was designed based on experimental mutagenesis data published on the CDK8–CycC complex and the CDK4–CycD1 complex. A R55A-E56A double point mutation in the αC-helix of CDK4, corresponding to R65A-E66A in CDK8, decreased its binding activity toward cyclin D1 by 85% [29]. On the basis of these results, Barette et al. introduce the R65A-E66A double point mutation in CDK8 and find that similarly to CDK4, this double point mutation greatly affects the capacity of CDK8 to bind to CycC. However, for the formed complex, they find that CDK8^R65A-E66A^ is still able to stabilize the complex CDK8^R65A-E66A–^CycC [28]. We therefore calculated the binding energies for the different CDK8–CycC complexes (systems 1a, 1b, 3, 5, 6, and 8) and summarized the results in Table 2.

Only the enthalpy part of the binding energy was calculated here. Indeed, the relative contribution of the entropic term to the ΔΔG is considered to be negligible when comparing two similar systems, such as, for example, in mutational studies, or when comparing ligands that bind to the same binding site (as is the case here), as both contributions are supposed to cancel each other out [30]. Therefore, in this study, ΔG corresponds to the binding free energy without the entropic term. In agreement with our structural and dynamical observations, the binding free energy values range from −141.0 ± 0.2 to −124.6 ± 0.2 kcal·mol^−1^, which confirms the stabilizing effect of CycC. In particular, the result for the mutated system CDK8^R65A-E66A^ is consistent with the experimental observations, which report that the double point mutation does not affect the stabilization of the complex. The non-polar part of the free energy, composed of the Van der Waals term in the gas phase (ΔE_VDW_) and the non-polar part of the solvation energy term (ΔE_np_), is the major favorable component of the CycC binding. Its value is between −171.5 kcal·mol^−1^ and −200.7 kcal·mol^−1^ depending on the system. The highly favorable non-polar part of the free energy might come from the desolvation of the non-polar groups at the binding interface between CDK8 and CycC, as well as the hydrophobic interactions formed between the two partners. Such a phenomenon has been seen in several protein–protein interactions, where the main interactions that are responsible for the binding of proteins are hydrophobic in nature [31,32]. On the other hand, the very favorable electrostatic term in the gas phase (ΔE_eel_) is completely compensated by the unfavorable contribution of the polar part of the solvation free energy (ΔE_GB_), resulting in an unfavorable total electrostatics interaction between 40.0 kcal·mol^−1^ and 69.5 kcal·mol^−1^ depending on the system. This compensation phenomenon due to the desolvation penalty of polar groups upon complex formation has been discussed in several studies of protein–protein interactions [33].

#### 2.2.2. CDK8–CycC Binding Free Energy: Decomposition per Residue

The method of per-residue binding free energy decomposition can reveal the energy contribution of key residues involved in the protein–protein interaction interface. The total of 9500 snapshots extracted from the trajectories in the range of 50 ns^−1^ μs (i.e., 1 snapshot every 100 ps) was decomposed using the MM-GBSA method. We first identify the common list of residues that significantly contribute to the CDK8–CycC binding in all of the studied complexes (systems 1a, 1b, 3, 5, 6, and 8).

##### Hotspots Common to All Studied CDK8–CycC Complexes

For each of these systems, the important CDK8–CycC binding residues were extracted using the following condition as the cut-off: the absolute value of ΔG_total_ of the residue has to be superior to 1 kcal·mol^−1^. In the supporting information, the list of the extracted important residues of each system is represented as a barplot (Appendix A). To extract the common list of important residues shared by all of the studied complexes, we took the intersection of these different lists. The heat map presented in Figure 4 contains the common list of important residues (26 in total) and their binding free energy contributions.

The first obvious result is that no great difference in free energy values is seen between the different studied systems. Second, all of the residues present a favorable contribution to CDK8–CycC binding. Moreover, the 26 residues are uniformly distributed on the interaction surface. These first observations suggest that the studied complexes share a large and similar surface of interaction.

##### Hotspots Common to the CDK Family

The members of the human CDK family share a conserved common interaction surface with their Cyc partner. This common interaction surface includes the β3–αC region, the αC-helix, and the post-αC region (β4–β5) of the CDK protein in contact with the H5-helix, the H5-H1′ loop, and the residues on both sides of the H3-helix of the Cyc [34,35]. In total, 73.1% of the identified common important residues of the CDK8–CycC interaction belong to this conserved core, as we can see on the heat map (Figure 4). We subsequently analyzed the interactions between CDK8 and CycC involving these common important residues of the CDK family’s conserved core.

This conserved core is located at the center of the interaction surface, and it is mainly composed of hydrophobic residues. Among them, the central Phe140^CycC^ situated on the CycC H5-helix seems to have a crucial role in CDK8–CycC binding. Indeed, in a parallel stacking, Phe140^CycC^ establishes a cation–π interaction with Arg91^CDK8^ of the β4–β5 loop, both characterized by a high ΔG absolute value (Figure 4). This interaction has an average occupancy of about 78.8% ± 8.3 along all of the simulations. It has been reported in the literature that a planar cation–π stacking between an arginine and an aromatic side chain may be a critical interaction for the function of a protein, including, in particular, in allowing the arginine to form other hydrogen bonds [36]. This is precisely the case here, as Arg91^CDK8^ also establishes a hydrogen bond with Glu137^CycC^ in the H5-helix with occupancy of 75.6% ± 12.2. Another residue of the CycC H5-helix, the Leu143^CycC^, is involved in a hydrophobic contact with Cys64^CDK8^ of the αC-helix with occupancy of 55.7% ± 13.0. Concerning the H5-H1′ loop, a hydrogen bond is formed between Cys148^CycC^ and Arg71^CDK8^, with occupancy of 87% ± 7.8, and a water bridge is formed between Ile151^CycC^ and Glu72^CDK8^ of the αC-helix, with occupancy of 74% ± 13.3%. Finally, in the C-terminus of the H3-helix, Lys96^CycC^ interacts with Ile59^CDK8^ localized in the β3–αC loop through a hydrogen bond with occupancy of 92.4% ± 7.6%.

In summary, the studied complexes (systems 1a, 1b, 3, 5, 6, and 8) display a large common binding surface composed of 26 residues distributed uniformly along the interaction surface. This common binding surface is also very similar because the free energy values present few variations from one system to another. All of the 26 residues contributed favorably to CDK8–CycC binding, with free energy values ranging from −9.4 kcal·mol^−1^ to −1.0 kcal·mol^−1^. In total, 73.1% of those residues (19/26 residues) belong to the conserved common interaction interface in the human CDK/Cyc family. Interestingly, we found that the remaining nine residues belong to regions that are specific to CDK8.

##### Hotspots Specific to CDK8–CycC

Involving the N-terminus segment of CycC

Although the cyclins are less similar in sequence among themselves compared with the CDKs, they share a common fold constituted of two cyclin boxes comprising five helices each (H1-H5 and H1′-H5′), which are generally associated with two additional helices at the N-terminus and the C-terminus segments, noted as H_N_ and H_C_, respectively (Figure 5). Unlike cell cycle cyclins (cyclin A/B/D/E), in transcriptional cyclins (cyclin C/T/K/H) [35], the H_N_ is located on the side opposite to the CDK binding surface, and it is not involved in kinase recognition. However, in this case, the N-terminus of CycT is still able to maintain some contacts with CDK9. CDK8–CycC appears as an exception, because the CycC N-terminus segment is part of the interaction surface positioned below the αC-helix and between the CDK8 αE-helix and the CycC H5-H1′ loop (Figure 4). A strong hydrogen bond interaction is observed between the Glu72^CDK8^ and the Ser9^CycC^, with occupancy of 87 ± 9.3% along all of the simulations.

Involving the CDK8-specific N-terminus helix (αB-helix)

CDK8 exhibits an additional N-terminus αB-helix (residues 1-12) preceding the αC-helix, which is unique within human CDK family members [15]. Other CDKs display a shorter N-terminus segment of 5–10 residues, except CDK9, where the segment is of equal length but unstructured (random coil). Among the identified common important binding residues (Figure 4), many of them interact with the αB-helix. In particular, we observed interactions between the proline rich C-terminus segment of CycC and the αB-helix. The Pro260^CycC^ and the Ser80^CycC^ both establish a hydrogen bond with Asp2^CDK8^, with an average occupancy of 82.1% ± 10.3 and 79.1% ± 10.4, respectively. The Lys261^CycC^ interacts with Tyr3 ^CDK8^ and Asp4^CDK8^, with an average occupancy of 83.4% ± 9.9 and 73.3% ± 11.1, respectively. The CDK8 αB-helix also forms a hydrophobic interaction, particularly the Leu9^CDK8^ with Phe140^CycC^, with an occupancy of 88.2% ± 7.5.

Taking these results together, it appears that strong and favorable interactions are formed between the proline-rich C-terminus segment, which shows a dramatic divergence in length and orientation among CycC partners, and the CDK8-specific αB-helix. Together with the contacts involving the N-terminus segment of CycC, these strong interactions are specific to the CDK8–CycC complex and could be one of the mechanisms explaining the selectivity of CDK8 against CycC. Indeed, unlike CDK2, which can bind different Cyc partners (Cyc A/B/E) [37], CDK8 is specific to CycC. Moreover, experimental mutational studies converge with our observations as the mutant CDK8–CycC complex missing the αB-helix (the first 22 residues in the N-terminus segment of CDK8) has an affinity of 300.71 nM against 7.05 nM for the native complex [15]. Thus, in addition to mediating a specific interaction between the CDK8 and CycC, the αB-helix also contributes to ensuring tight binding between CDK8 and CycC. For comparison, the affinities of native CDK9–CycT1, CDK2–CycA, and CDK7–CycH are weaker by at least one order of magnitude at 300 nM [38], 52 nM, and 57 nM, respectively [39]. It is generally assumed that a high affinity to a partner compared to other homologous partners leads to highly specific binding to the considered partner. This may be achieved through small structural variations, which seem to occur here, in the CDK8–CycC’s recognition of the αB-helix. Targeting the highlighted specific interaction hotspots between CDK8 and CycC could be a promising approach to designing a peptide that specifically inhibits the CDK8–CycC activity by preventing the binding of CDK8 to CycC. Two peptides targeting the CDK2–CycA interface were reported, but neither of them has yet made it to the clinic. The first one binds at the core of the common binding surface at the αC-helix/H5-helix interface [40]. The second one targets a surface pocket in CycA, which is a structurally conserved domain comprising the H3, H4, and H5 helix of cyclin A [41].

##### Difference in Binding Surface between the Different Complexes

After deciphering the common molecular features of the CDK8–CycC interaction surface, we now want to assess whether a significant difference exists between the binding surfaces of the studied complexes. In order to highlight possible differences in energy contributions of the residues, we extract the list of the residues that form at least one significant interaction (using the same cut-off as above, absolute (ΔG) > 1 kcal·mol^−1^) in one of the studied complexes. In other words, instead of taking the intersection of the lists of important residues of each system, as we did previously to obtain the common molecular features, we took the union of these lists. The resulted matrix has been attached in the supporting information (Appendix A). To compare the contributions of the residues of each system with each other in a convenient way, we calculated a correlation matrix from the contribution matrix, and we present the results as a scatterplot matrix (Figure 5).

##### *DMG-out* CDK8–CycC Complexes

The residues of *DMG-out* conformation complexes (1a, 1b, 3, 5, and 6) display very similar energy contribution values as the correlation coefficients are between 0.74 and 0.90. The mutated *DMG-out* conformation complex (system 5) does not exhibit a significant difference from the other native *DMG-out* conformation complexes (1a, 1b, 3, and 6) in terms of the energy contribution of residues. Indeed, it presents a correlation coefficient always superior to 0.74 against them. It is particularly close to system 6 (correlation coefficient = 0.88). Moreover, the double point mutation (CDK8^R65A_ E66A^) does not significantly affect the binding interaction network between CDK8 and CycC. Therefore, the mutant complex presents a similar stability (Table 2) associated with a similar binding interaction network compared to native systems. Together, these results indicate that the *DMG-out* conformation complexes share a similar binding interaction surface.

##### Difference in Binding Surface between *DMG-in* and *DMG-out*

Although the studied CDK8–CycC complexes share a large common interaction surface, as we detailed previously, the distribution of the energy contribution of the residues of the *DMG-in* complex is the least correlated with that of the other complexes, with a correlation coefficient between 0.51 and 0.63. In the *DMG-in* conformation complex, the CycC is slightly shifted toward CDK8, as shown in Figure 6. This shift increases the contacts between the CycC H3-H4 loop and the CDK8 activation loop, which is folded toward the CycC in the *DMG-in* conformation.

As a consequence, Arg178^CDK8^ and Pro183^CDK8^ of the activation loop that did not contribute to CDK8–CycC binding in the *DMG-out* conformation complexes are now close to CycC and present a favorable contribution (Appendix A). Moreover, the shift of the CycC modifies the interaction network at the CDK8–CycC interface, which might explain, for some residues, a change in their energy contribution, including Arg13^CDK8^ of the αB-helix, Leu86^CDK8^ on the β4 strand, Asn145^CDK8^, Trp146^CDK8^ at the C-terminus of the αE-helix, Ala2^CycC^ and Gly3^CycC^ of the N-terminus segment, and three residues at the N-terminus of the CycC H3-helix (Ile81, Asp82, and leu85) (Appendix A). Interestingly, other residues that are far from the interaction surface but part of the binding site also display a difference in their energy contribution in the *DMG-in* conformation system compared to the *DMG-out* one, including the Val27^CDK8^ and the Val35^CDK8^, which are part of the P-loop, Tyr99^CDK8^ and Ala100^CDK8^ in the hinge region, and Arg356^CDK8^ of the C-terminus of CDK8.

### 2.3. Activation Mechanism of CDK8

In the *DMG-in* conformation complex, we observe that the shift of the CycC toward CDK8 allows the Glu99^CycC^ to be closer to Arg65^CDK8^. Glu99^CycC^ establishes hydrogen bonds with Arg65^CDK8^, Arg178^CDK8^, and, to a lesser extent, Arg150^CDK8^. The three arginines also interact with each other through water-mediated hydrogen bonds. This interaction network is maintained over time (Appendix A) and could therefore have a role in the stabilization of the activation loop in the *DMG-in* conformation. In this context, we turn to the literature to find a possible known role of Arg65^CDK8^, Arg150^CDK8^, and Arg178^CDK8^ in the activation mechanism of CDK8. In that regard, it was reported that these three arginines are conserved within human CDK members, and they are involved in the second step of the activation mechanism [22]. As mentioned earlier in the introduction, the second step of the general activation mechanism of CDKs is the phosphorylation of a residue within the activation loop. The phospho-residue serves as an anchor to adjust the orientation of three conserved arginines, thereby inducing a *DMG-in* conformation of the activation loop. In CDK8, these three conserved arginines are Arg65^CDK8^, Arg150^CDK8^, and Arg178^CDK8^. However, because in CDK8 there is no phosphorylation, on the basis of crystallographic structure analysis, Glu99^CycC^ was hypothesized to mimic the missing phospho-residue within CDK8, and it serves as anchor to adjust the orientation of the three important arginines, Arg65^CDK8^, Arg150^CDK8^, and Arg178^CDK8^, in CDK8 [22]. The stable interaction network formed by the three arginines and Glu99^CycC^ observed during the MD simulation supports this hypothesis.

To further investigate this hypothesis and to achieve a dynamic view of the process, we simulate through targeted molecular dynamic simulation the conformational transition from a *DMG-out* conformation complex to a *DMG-in* one. The restraint was applied only on the activation loop (and not on the whole complex) because we want to verify if a relationship exists between the shift of the CycC and the conformational change of the activation loop (residue 171 to 182). We first check the stability of the protein structure over time during the TMD simulation by verifying the RMSF, the RMSD of the protein, and the restraint potential over time (Appendix A). The *DMG-in* conformation obtained through TMD simulation followed by 50 ns of cMD is in agreement with that of system 8 (Appendix A).

To monitor the shift of the CycC toward CDK8, we measure the distance between Glu99^CycC^ and a stable residue of CDK8, the Lys153^CDK8^ (according to its RMSF, cf. Figure 1). As the activation loop gets closer to the CycC, the CycC shifts toward CDK8, as shown by the Lys153^CDK8–^Glu99^CycC^ distance curve over time (Figure 7a). At the beginning of the TMD simulation, Arg178^CDK8^ first interacts with the Arg150^CDK8^ (Figure 7a), and, at this stage, the CycC already undergoes a small shift. This displacement of the CycC enables the Glu99^CycC^ to become closer to Arg65^CDK8^ and optimize its interaction with it. Then, we observe an interaction of Arg178^CDK8^ with Glu99^CDK8^, thus breaking the interaction between Arg178^CDK8^ and Arg150^CDK8^. The gradual rapprochement of the CycC toward CDK8 during the 50 ns of cMD production allows it to reform the interaction between the two arginines. Therefore, the displacement of the CycC might be an important event to adjust the orientation of the three conserved arginine residues. During these 50 ns of cMD production, the Glu99^CycC^-mediated hydrogen bond interaction network stabilizes, and a similar interaction network to that in system 8 is formed at the end (Appendix A). Arg178^CDK8^ becomes sandwiched between Arg150^CDK8^ and Arg65^CDK8^, and a hydrogen bond network is formed by the three arginines and Glu99^CycC^ (Figure 7). The three arginines interact with each other through water-mediated hydrogen bonds. It may be noted that finding this network is not trivial, as only the activation loop (residues 171 to 182) and, therefore, only Arg178^CDK8^ were submitted to the restraint potential (Arg150^CDK8^, Arg65^CDK8^, and Glu99^CycC^ were not under restraint).

From these results, it appears that the Glu99^CycC^ and the shift of the CycC are important for orienting and stabilizing the three conserved arginines known to be involved in the second step of the general activation mechanism of other CDK members. Therefore, our observations support the hypothesis that the Glu99^CycC^ in CDK8 mimics the missing phospho-residue, whose role is to adjust the orientation of three conserved arginines, thereby inducing a *DFG-in* conformation of the activation loop. In addition to that, our results suggest that a shift of the CycC toward the CDK8 is also required to obtain the active form of CDK8–CycC.

## 3. Materials and Methods

### 3.1. Material Description

The catalytic site of CDK8 lies between the N- and C-terminal lobes, as in other kinase proteins. Two conformations of CDK8 exist in the PDB that are differentiated by the conformation of the activation loop, which adopts either a *DMG-in* or a *DMG-out* conformation. CycC interacts mainly with the N-terminal lobe (Figure 8). The studied systems are summarized in Table 3. The corresponding crystallographic structures all come from the paper by Schneider et al. [24]. The structure 4F6U (PDB ID) presents the best resolution among all *DMG-out* structures resolved up to now. This structure is co-crystallized with a type II inhibitor (system 1a and 1b). To be sure that the results obtained are not ligand-dependent, the apo form of 4F6U (system 3) and another type II inhibitor (PDB ID: 4F7L) (system 6) with a slightly different binding mode (Appendix A) were also simulated. Then, to compare our results with experimental mutagenesis results, two residues of the αC-helix were mutated in the structure 4F6U (system 5). Finally, a *DMG-in* conformation of the complex (PDB ID: 4F7S), which is the conformation usually observed in the presence of CycC, was also simulated in order to compare the behavior of CycC in the complexes of the *DMG-in* and *DMG-out* conformations (system 8). These systems were also modeled without the CycC in order to investigate the effect of the CycC (except system 5).

### 3.2. Model Building

The structure of PDB ID 4F6U presents 3 missing loops: the activation loop containing the key DMG-motif (residues 177 to 193) and the loops from residues 116 to 120 and residues 240 to 244. In order to reconstruct these missing residues, we aligned the UniProt [42] canonical sequence of CDK8 on the PDB database to retrieve the most homologous template structures with the missing regions resolved and the activation loop in the *DMG-out* conformation. Two crystallographic structures of the human homologous CDK6 (PDB ID: 1BI8 and 1G3N) were retained and used as template structures. The sequence alignment was performed with Clustal Omega [43] with a particular focus on the alignment of domain kinase conserved motifs. CDK6 shares 37% of its identity and 63% similarity with CDK8 (Appendix A). Only missing regions in the target structure were rebuilt in order to keep the coordinates of the resolved parts of the protein unchanged. The sequence of CycC and the information regarding the presence of crystallographic molecules of water and a ligand (ligand ID 0SR) were conserved during the modelling of the missing part of CDK8. Finally, MODELLER version 9.16 [44] was used in order to generate the model. We thus obtained a model of CDK8 (residues 1 to 359) complexed to CycC (residues 1 to 264) and the ligand. The missing C-terminus segments of CDK8 (residues 360 to 464) and of CycC protein (residues 265 to 283) were not reconstructed. The complete model was subjected to structural validation through PROCHECK [45] and ProSA-web tools [46] (Appendix A). We did not build another model for the structure of PDB ID 4F7L but rather derived the model by replacing the inhibitor 0SO in that model (chemical replacement). Chemical replacement was considered sufficient because the orientation of the binding site residues is highly conserved in the two structures (PDB ID: 4F6U and 4F7L) and their respective inhibitors (ligand ID: 0SR and 0SO) share the same scaffold bound in the same orientation within the binding site (Appendix A). Therefore, the full structure of CDK8–CycC complexed with the ligand 0SO was obtained by first aligning the crystallographic structure 4F7L to the model and then by placing the ligand and the crystallographic molecules of water inside. We manually adjusted some residues to be in agreement with protein–ligand interactions observed in the crystallographic structure of PDB ID 4F7L using Molecular Operating Environment (MOE) version 2016.0802 from the Chemical Computing Group. The same procedure as the one described above was followed to fill the 3 missing loops of the structure of PDB ID 4F7S (which are the activation loop residues from 187 to 195 and the loops from residues 116 to 121 and from residues 238 to 242). The crystallographic structures of the human homologous CDK1 (PDB ID 1P5E) and CDK2 (PDB ID 1P5E) were retained and used as template structures. CDK1 and CDK2 share, respectively, 37.8% and 38.3% of their identity and 54.5% and 55.9% similarity with CDK8.

### 3.3. System Preparation

In total, 9 systems were prepared (all described in Table 3). The AmberTools 14 suite [47] was employed to protonate, solvate, neutralize, and generate the topology and coordinate the files of the systems. Ligands were prepared by using the Antechamber tool and the GAFF force-field after adding hydrogen atoms with the reduce utility [48,49]. The three inhibitors were modeled in their neutral state. Further analysis was carried out for the protonation state of the inhibitor 0SR (ligand ID) (Appendix A), as the pKa of alkylmorpholines is about 7.4 [50]. The morpholine of the inhibitor 0SR was finally modeled in its unprotonated state. Partial charges on the ligands were generated through the AM1/BCC method [51]. PROPKA version 3.0 [52] was used to check the protonation state of ionizable residue side-chains at pH = 7. The protein force-field ff14SB parameters were assigned [53]. Then, the system was solvated in a rectangular TIP3P water box, with the side of the box being at least 10 Å away from any solute atom. Finally, Cl^-^ ions were added to neutralize the positively charged system for a total number of atoms around 110,000 atoms.

### 3.4. Conventional MD Simulation (cMD)

A four-cycle minimization was performed with 2000 steps each cycle, minimizing first the solvent, second the residue side-chains, then the solute, and, finally, the entire system. The SHAKE algorithm [54] was applied to constrain bonds involving hydrogen atoms, allowing a time increment of 2 fs. Temperature regulation at 300 K was ensured through Langevin dynamics with a collision frequency of 2 ps^−1^. The long-range electrostatic interactions were computed using the Particle Mesh Ewald (PME) method beyond 10 Å distance. The system was slowly heated in canonical ensemble (NVT) from 0 to 300 K over a period of 50 ps, where a harmonic restraint on the solute (20 kcal·mol^−1^·Å^−1^ force-field constant) prevents the system from structural distortion. The system was then equilibrated during a 10 ns MD simulation in the isobaric–isothermal ensemble (NPT) at 300 K and 1 atm, through which the harmonic restraint was gradually decreased from 20 kcal·mol^−1^·Å^−1^ to 3 kcal·mol^−1^·Å^−1^ in 1.3 ns and then totally relaxed during 8.7 ns. The pressure relaxation time was set to 1 ps. cMD calculations were performed using the PMEMD.cuda module of the AMBER14 program [47]. We performed 1 μs of cMD production on each system presented in Table 3 and saved the coordinate every 10 ps.

### 3.5. Targeted Molecular Dynamics (TMD)

The TMD is a simulation technique for determining the pathway of a conformational transition between two states: (un)bound, (un)folded, open/close conformation, etc. [55]. It consists of constraining the root mean square deviation (RMSD) between the current structure (which is the starting structure at the beginning of the simulation) and a reference structure (RMSD_current_) to a user-defined value, namely the RMSD_target_. This value of RMSD_target_ is slowly varied from an initial value to a targeted final value (RMSD_target_final_), which results in the simulation of the process leading to the final desired state. In the AMBER14 program, a harmonic restraining potential (V_restraint_) is added to the force-field to help the RMSD_current_ in reaching the successive values of RMSD_target_ until the final value (RMSD_target_final_).
(1)$$V_{restraint}=\frac{1}{2}\times f{\times N}_{atoms}{\times \left({RMSD}_{current}-{RMSD}_{target}\right)}^{2},$$
where f is the harmonic force constant and N_atoms_ is the number of restrained atoms, that is, the number of atoms for which the RMSD is calculated. Note that the atomic coordinates are mass-weighted in the calculation of RMSD. There exist two approaches for TMD: direct TMD and reverse TMD (TMD^−1^). We applied direct TMD. In direct TMD, the reference structure corresponds to the final targeted structure, so that the value of RMSD_target_ is decreased from the RMSD between the initial and the target structure to a value close to 0. In this study, the initial structure is the complex CDK8–CycC in the *DMG-out* conformation, and the target structure is that in the *DMG-in* conformation. The RMSD is calculated for the residues 171 to 182 of the activation loop, after aligning the current and the target structure on the backbone of the less flexible residues of the active site (90 residues in total: residues 26 to 105 and 148 to 158). The spring constant f was set to 2 kcal·mol^−1^. The RMSD_target_ was changed in increments of 0.12 Å every 50 ps from the value of 12.3 Å to 0.01 Å during a total simulation time of 5 ns. TMD runs were performed with the parallelized version of the SANDER module from the AMBER14 program. The TMD simulation was then continued by 50 ns of cMD simulation following the same parameters as described above.

### 3.6. RMSD, RMSF

The trajectories were aligned on the corresponding crystallographic structures using the heavy atoms of CDK8 as a mask. The root mean square deviation (RMSD) and the root mean square fluctuation were calculated using the same mask.

### 3.7. PCA

When applying MD simulations on biological systems, some questions are often raised. (i) Are the sampled conformations in one MD replicate similar to those extracted from a second replicate? (ii) Does the conformational sampling vary over time within the same trajectory? (iii) What are the protein regions whose movements contribute the most to explaining the conformational diversity? To answer such questions, the principal component analysis method (PCA) is a suitable method. PCA is a linear dimensionality reduction technique that linearly combines a set of variables (here, the coordinates of CDK8 backbone residues) into a reduced number of uncorrelated variables called principal components (PCs). The PCs correspond to the directions of largest variance, that is, the largest-amplitude fluctuations. To obtain the PCs, we first extracted the CDK8 backbone of the last 500 ns of a trajectory by selecting 1 snapshot every 2.5 ns (200 snapshots in total). Trajectories of the system simulated in the absence and presence of CycC are concatenated, leading to a total of 400 snapshots. It is important to align the trajectories to be analyzed on a same referential. Then, a covariance matrix was calculated from the atomic coordinate matrix of the trajectory. The eigenvectors of the covariance matrix are the PCs. The PCs were ordered with PC1, the direction of largest variance, PC2, the direction of second largest variance, etc. To visualize the largest amplitude motions, a PDB format trajectory was produced that interpolates between the most dissimilar structures in the distribution along PC1. PCA analyses were performed with bio3d package [56].

### 3.8. MM-GBSA

The molecular mechanics generalized Born surface area (MM-GBSA) method supplied with AMBER were used to calculate the protein–protein free energy [57]. In total, 9500 snapshots were extracted from the trajectories in the range of 50 ns^−1^μs (i.e., 1 snapshot every 100 ps). The binding free energy is calculated as follows:(2)$${\Delta G}_{bind}=\langle G_{complex}\rangle -\langle G_{receptor}\rangle -\langle G_{ligand}\rangle ,$$
where $\langle G_{x}\rangle$ corresponds to the average of the total free energy of the component x over snapshots taken from the MD trajectory. The total free energy of each molecule is computed from the following equation:(3)$$G=E_{MM}+G_{sol}–TS,$$
where $E_{MM}$ is the molecular mechanical energy, $G_{sol}$ is the solvation free energy, and the term TS is the entropic contribution. The solvation free energy is the sum of the polar and non-polar contributions. The non-polar contribution is attributed to cavity formation in the solvent and van der Waals interactions between the solute and the solvent, which are typically calculated from the solvent-accessible surface area. The polar contribution of $G_{sol}$ is obtained following the generalized Born model [58] available in AMBER.

While the molecular mechanics energy term can be easily obtained from the results of a molecular dynamics simulation, the entropic term is often difficult to achieve. It can be approximated through a quasi-harmonic approximation or calculated through a normal mode analysis. However, the calculation is time-consuming, and it can be affected by large errors. Such a calculation was not considered in this study.

### 3.9. Other Analysis Tools

The VMD program [59] and the CPPTRAJ module from the AMBER14 program [47] were also used to manipulate and analyze trajectories. The analysis of the protein–protein interactions was performed with the Structure Interaction Diagram module of the maestro suite [60].

## 4. Conclusions

Theoretical studies were conducted on the human CDK8–CycC complex in order to gain a greater understanding of the binding of CycC to CDK8, which is an important target in cancer therapy. We first investigated the role of CycC in the structure and dynamics of CDK8. We found that the CycC is crucial for maintaining the proper structure and dynamics of CDK8 in both the active (*DMG-in*) and inactive (*DMG-out*) forms of the complex. Unlike CDK2, where the binding of a type II inhibitor to CDK2–CycB results in the dissociation of CycB from CDK2 in a competitive manner [27], Schneider et al. have shown that the binding of a type II inhibitor to CDK8–CycC does not dissociate CycC [24]. Our findings agree with this result as the presence of a type II inhibitor does not affect the stabilizing effect of the CycC toward CDK8. The free energy values of CDK8–CycC binding calculated through the MM-GBSA method confirm these results and show that the CycC stabilizes both CDK8 forms (active and inactive) to the same extent.

The analysis of the interaction between CDK8 and CycC, through the method of per-residue binding free energy decomposition, highlighted 26 hotspot residues uniformly distributed on the interaction surface that strongly and favorably (ΔG_total_ < −1 kcal·mol^−1^) contribute to CDK8–CycC binding in all studied CDK8–CycC complexes. In total, 19 of the 26 important residues belong to the conserved common interaction surface in the human CDK family. On the contrary, the remaining seven hotspot residues are situated in two binding sites of the interaction surface that are specific to the CDK8–CycC complex and involve the proline rich C-terminus segment, the CDK8 αB-helix, and the N-terminus segment of CycC. These key amino acids proposed in this work provide valuable information to design an inhibitor that will effectively prevent the binding of the CycC to CDK8, which will block the activation of the complex, thereby interfering with the function of CDK8 as an oncogene. The active and the inactive forms display some differences in their CDK8–CycC binding energy contribution values. These differences might be explained by the flip of the activation loop from a *DMG-out* to a *DMG-in* conformation and the displacement of the CycC toward CDK8 in the active form.

The simulation of the conformational transition from the inactive to the active form through TMD simulation showed that this displacement of the CycC toward CDK8 occurs during the conformational change. This displacement is an important event to adjust the orientation of three conserved arginine residues (Arg65^CDK8^, Arg178^CDK8^, and Arg150^CDK8^), which is meditated by the Glu99^CycC^, thereby inducing a *DMG-in* conformation (active form). The active form is maintained through a hydrogen bond interaction network involving the three arginines and the Glu99^CycC^. In the human CDK family, the three conserved arginine residues, together with a phosphorylated residue, are known to have a role in the conformational change of CDK and the stabilization of the active form. Our TMD simulation suggests that Glu99^CycC^ assumes the role of the missing phosphorylated residue in CDK8.

Our study provides interesting molecular insights, describing the interaction between CDK8 and CycC in terms of structure and energy. Because this interaction is essential to the activity of CDK8, the particular characteristics of this interaction and its mechanism of activation highlighted in this study are valuable information for designing specific compounds targeting the CDK8/CycC interface. In a more general view, these results point to the importance of keeping the CycC in computational studies when studying the human CDK8 protein in both the inactive and active forms.

## Figures and Tables

**Figure 1 ijms-25-05411-f001:**
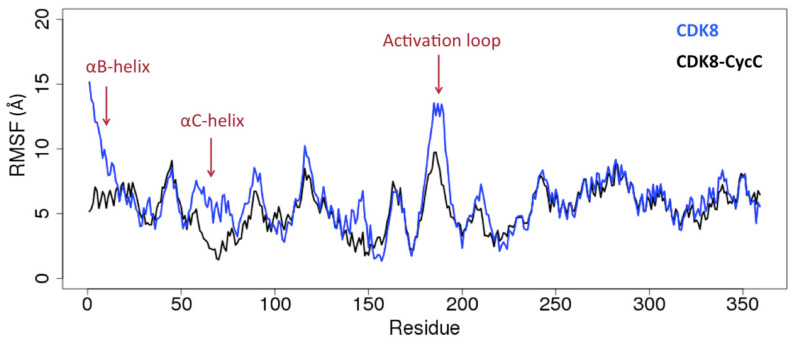
Root mean square fluctuations (RMSF) per residue of CDK8 in presence (black) and absence (blue) of CycC during 1 μs of simulation.

**Figure 2 ijms-25-05411-f002:**
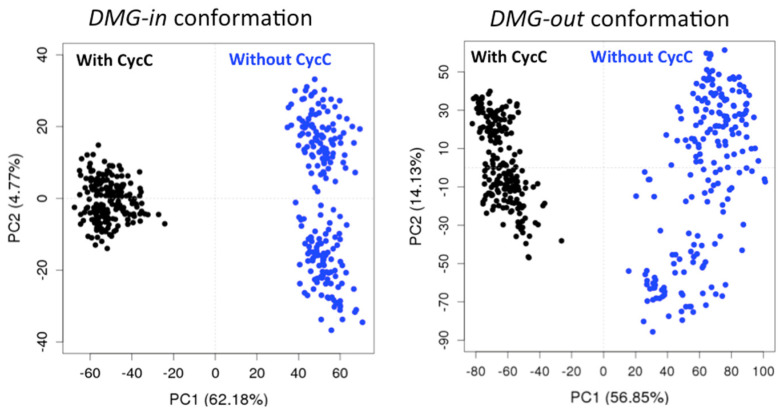
PCA projection of the structural evolution of CDK8 in *DMG*-*in* conformation (systems 8/9, left) and *DMG*-*out* conformation (systems 1a/2a, right) in presence (black) and absence (blue) of the CycC during the MD simulations. One snapshot of a trajectory is represented by a dot in the individual map of PC1 against PC2.

**Figure 3 ijms-25-05411-f003:**
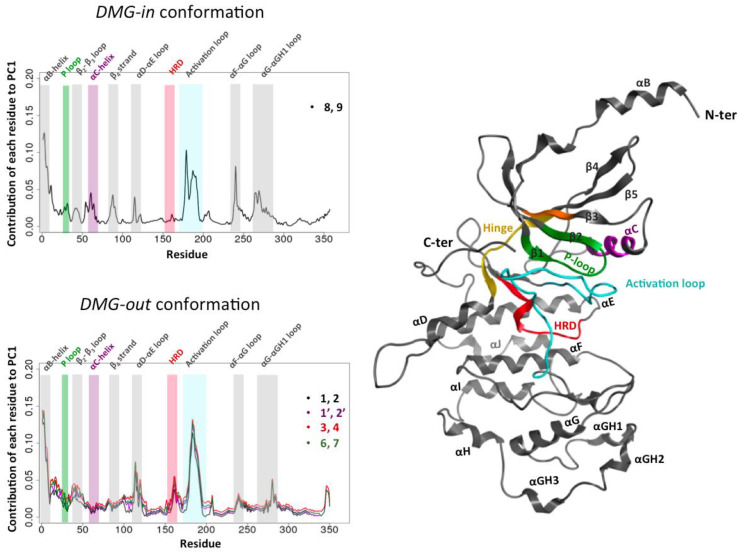
Impact of the CycC on the structure of CDK8: contribution of each residue of CDK8 to PC1. Loading plot of the PCA performed on the combined trajectory (with/without CycC) in the *DMG-in* conformation system (top left) and *DMG-out* conformation systems (bottom left). On the right is a representation of CDK8 using a dark gray ribbon, except the regions of the kinase domain containing the conserved motifs.

**Figure 4 ijms-25-05411-f004:**
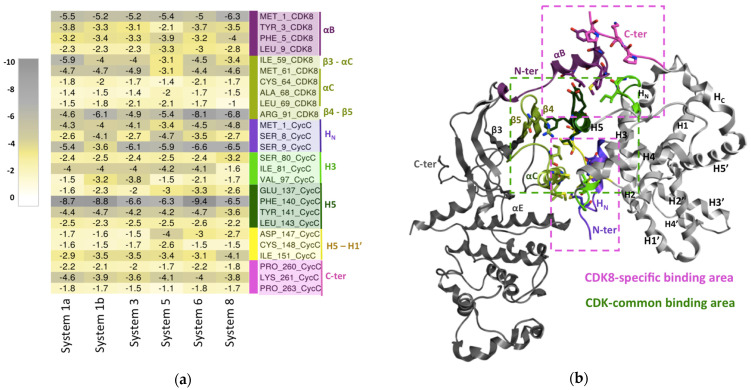
CDK8–CycC binding interface. (**a**) Matrix of the per-residue energy contribution (ΔG without entropy). The residues are those that highly contribute to CDK8–CycC binding (absolute (ΔG) > 1 kcal·mol^−1^) in all studied CDK8–CycC complexes. Residues are tagged according to the secondary structure they belong to. The residues colored in pink–purple tones are those belonging to CDK8-specific binding sites. Those colored in green–yellow tones are those belonging to human CDK common binding sites. (**b**) CDK8/CycC structure and the binding sites at the interface of CDK8–CycC. CDK8 is represented by the dark gray ribbon, except the secondary structures with residues that highly contribute to CDK8–CycC binding in all studied CDK8–CycC complexes. Idem for CycC, which is colored in light gray. Besides the common binding area (represented by a dashed green box), the CDK8/CycC complex forms additional contacts mediated by the CDK8 N-terminus αB-helix and the CycC N-terminus, including the HN helix (highlighted by dashed pink boxes).

**Figure 5 ijms-25-05411-f005:**
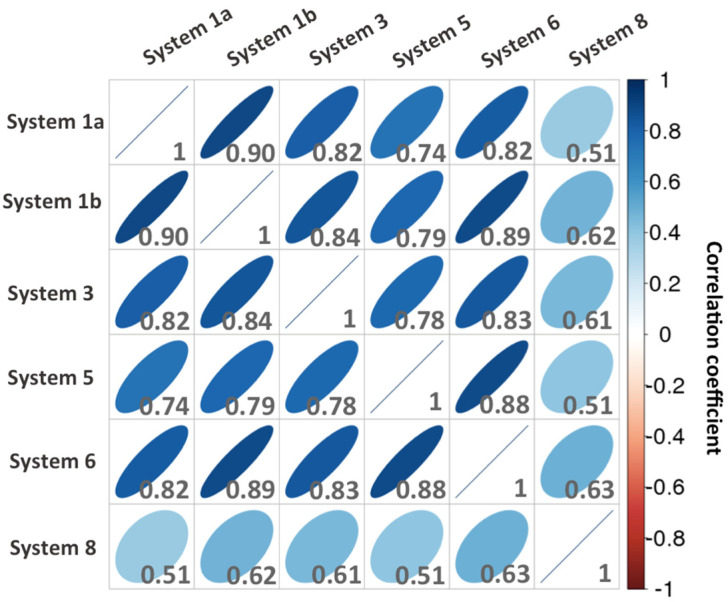
Correlation plot matrix of the residues’ energy contributions to CDK8–CycC binding of each system. The selected residues present at least one significant energy contribution (absolute (ΔG) > 1 kcal·mol^−1^) in one system.

**Figure 6 ijms-25-05411-f006:**
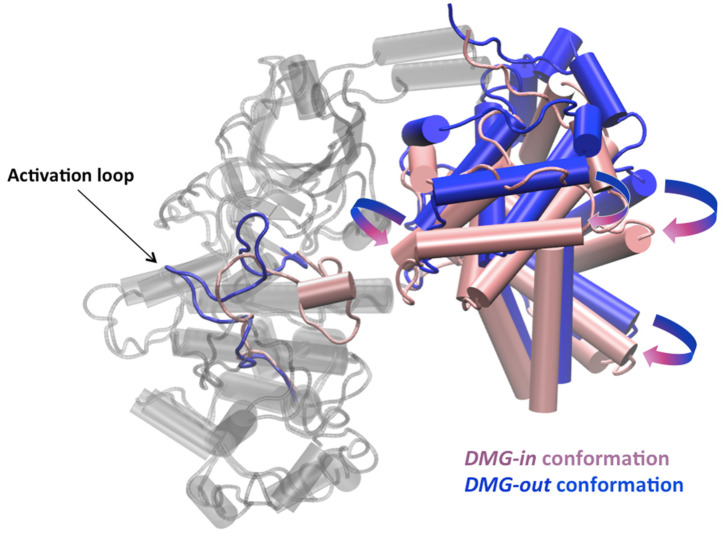
Pipes and planks representation of the *DMG-in* and *DMG-out* conformations of the CDK8–CycC complexes in a cartoon. CDK8 structures are colored in gray, except the activation loop. The activation loop and the CycC structures are colored in pink in the *DMG-in* conformation and in blue in the *DMG-out* conformation.

**Figure 7 ijms-25-05411-f007:**
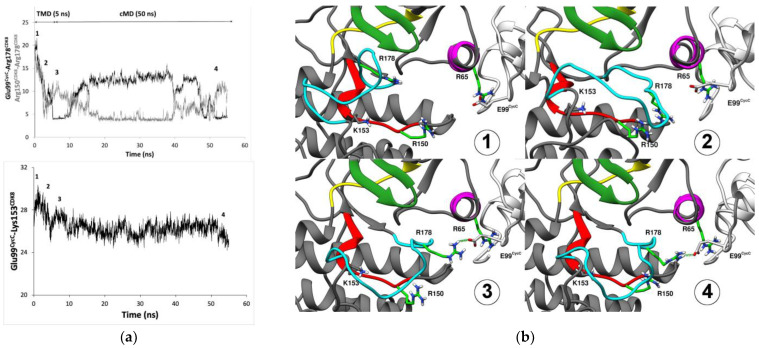
Conformational transition of the CDK8 activation loop from *DMG-out* to *DMG-in* conformation. (**a**) Plots of the measured distance between the pairs Glu99^CycC^ and Arg178^CDK8^, Arg150^CDK8^ and Arg178^CDK8^, and Glu99^CycC^/Lys153^CDK8^. The distances between Glu99^CycC^ and Arg178^CDK8^ (black line) and Arg150^CDK8^ and Arg178^CDK8^ (gray line) enable us to monitor the transition of the activation loop over the simulation time. The distance between Glu99^CycC^ and Lys153^CDK8^ enables us to monitor the displacement of the CycC toward CDK8. (**b**) Representation of the three conserved arginines, Arg65^CDK8^, Arg150^CDK8^, and Arg178^CDK8^, and Glu99^CycC^ over the simulation time course. The image numbers 1 to 4 correspond to their position along the trajectory reported on the plots. CDK8 is represented by the dark gray ribbon, except the regions of the kinase domain containing the conserved motifs; in particular, the activation loop is in cyan. CycC is in light gray. Arg65^CDK8^, Arg150^CDK8^, Arg178^CDK8^, Glu99^CycC^, and Lys153^CDK8^ are represented by sticks, arginines are light green, glutamate is light gray, and Lys153^CDK8^ is dark gray.

**Figure 8 ijms-25-05411-f008:**
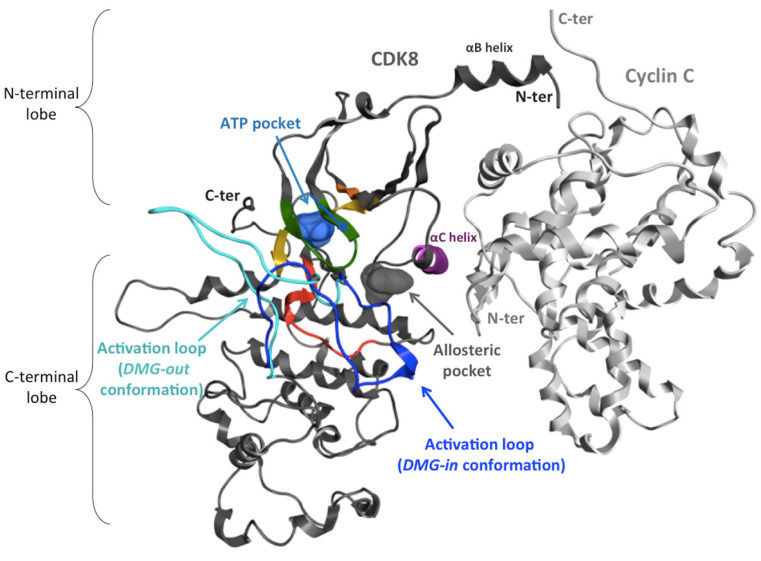
Ribbon representation of the CDK8–CycC complex. Cyclin C (CycC, PDB ID: 4F6U) is colored in light grey and CDK8 in dark gray, except the conserved motifs of the kinase domain. Among these motifs, the activation loop in the *DMG-out* conformation (inactive form) is colored in cyan (PDB ID: 4F6U), and that in the *DMG-in* conformation (active form) is dark blue (PDB ID: 4F7S).

**Table 1 ijms-25-05411-t001:** Comparison of the average RMSD calculated from the heavy atoms of CDK8 during 1 μs simulation for the systems with and without CycC. Systems 1a, 2a, 1b, 2b, 3, 4, 5, and 6 are in the *DMG-out* conformation, and systems 8 and 9 are in the *DMG-in* conformation.

System ID with Cyclin C	Average RMSD (Å) (±Standard Deviation)	System ID Without Cyclin C	Average RMSD (Å) (±Standard Deviation)
1a	3.6 ± 0.2	2a	5.5 ± 1
1b	3.9 ± 0.3	2b	5.5 ± 0.3
3	3.8 ± 0.3	4	5.4 ± 0.9
6	3.2 ± 0.2	7	5 ± 0.6
8	3.4 ± 0.3	9	3.7 ± 0.4

**Table 2 ijms-25-05411-t002:** The binding free energy and energy components of the CDK8–CycC complex calculated using the MM-GBSA method and averaged on the simulations. All of the energies are reported in kcal·mol^−1^, with their corresponding standard errors. ΔE_eel_ and ΔE_VDW_ are, respectively, electrostatic and van der Waals contributions in the gas phase. ΔE_GB_ and ΔE_np_ are, respectively, electrostatic and non-polar contributions in the solvation phase. ΔG_total_ is the total binding free energy without considering the entropic term.

Systems’ PDB ID Conformation	System 1a (4F6U) *DMG-out*	System 1b (4F6U-Replica) *DMG-out*	System 3 (4F6U-apo) *DMG-out*	System 5 (4F6U^R65A_E66A^) *DMG-out*	System 6 (4F7L) *DMG-out*	System 8 (4F7S) *DMG-in*
ΔE_VDW_	−163.0 ± 0.1	−160.7 ± 0.1	−160.0 ± 0.2	−148.7 ± 0.1	−156.3 ± 0.1	−176.5 ± 0.1
ΔE_eel_	−508.4 ± 1.0	−436.5 ± 1.0	−490.0 ± 1.5	−573.9 ± 1	−500.6 ± 0.9	−588.5 ± 0.9
ΔE_GB_	554.3 ± 0.9	491.1 ± 0.9	543.2 ± 1.4	613.9 ± 0.9	541.0 ± 0.8	613.9 ± 0.8
ΔE_np_	−23.9 ± 0.0	−23.1 ± 0.0	−23.2 ± 0.3	−22.7 ± 0.0	−22.9 ± 0.0	−25.4 ± 0.0
ΔG_total_ (Without entropy)	−141.0± 0.2	−129.2 ± 0.2	−130.0 ± 0.3	−131.5 ± 0.2	−138.8 ± 0.2	−124.6 ± 0.2

**Table 3 ijms-25-05411-t003:** Description of the studied systems.

System ID	PDB ID	Ligand Name	DMG Conformation	Manipulation
1a	4F6U	0SR	*DMG-out*	(-)
1b (replica)	4F6U	0SR	*DMG-out*	(-)
2a	4F6U	0SR	*DMG-out*	Removal of CycC
2b (replica)	4F6U	0SR	*DMG-out*	Removal of CycC
3	4F6U	(-)	*DMG-out*	Removal of ligand
4	4F6U	(-)	*DMG-out*	Removal of ligand and CycC
5	4F6U	0SR	*DMG-out*	CDK8 mutations: E66A, R65A
6	4F7L	0SO	*DMG-out*	(-)
7	4F7L	0SO	*DMG-out*	Removal of CycC
8	4F7S	0SW	*DMG-in*	(-)
9	4F7S	0SW	*DMG-in*	Removal of CycC

## Data Availability

All PDB structures used to build the initial models of MD simulations were downloaded from the RCSB protein data bank https://www.rcsb.org (accessed on 1 March 2024). Homology modeling was realized by using the Modeler software “https://salilab.org/modeller/” (accessed on 1 March 2024). The licensed Amber14 was used to perform MD simulations https://ambermd.org (accessed on 1 March 2024). The publicly available AmberTools15 was used to analyze the MD trajectories, in addition with VMD https://www.ks.uiuc.edu/Research/vmd/ (accessed on 1 March 2024). Data simulations are available upon request by mailing the corresponding author (S.A.S).

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
