# Peer review of "Highlighting the Major Role of Cyclin C in Cyclin-Dependent Kinase 8 Activity through Molecular Dynamics Simulations"

_ijms, 2024, doi:10.3390/ijms25105411_

Round 1

Reviewer 1 Report

Comments and Suggestions for Authors

The authors conducted a comprehensive study on the interaction between CDK 8 and cyclin C. They established that the presence of Cyclin C enhances the stability of CDK 8 and identified the key residues involved in this interaction. Using MD simulation, they demonstrated how the DMG-in or out states are associated with the relative position between CDK 8 and cyclin C, and highlighted the crucial role of E99CycC in the CDK8 activation mechanism. The authors presented their findings and explained the computational tools used clearly, making them accessible to readers without extensive computational background.

However, a few points need further clarification:

1. The authors address that Fig. 2 does not indicate the activation loop and the αB-helix adopts unique conformations in the absence of CycC. However, considering the fact that the activation loop and the αB-helix have dominant impacts on the PC1 value, we can consider PC1 as a featurization of their structure as a whole. Fig. 2 indicates that the presence or absence of CycC results in distinct PC1 values, suggesting a significant difference in the structure. Furthermore, Fig S8 seems to suggest that the presence of CycC leads to a notable change in the structure of the αB-helix, while the activation loop can adopt similar conformations. A clearer interpretation of Fig. 2 is needed.

2. During TMD simulations (Fig 7a), there are abrupt changes in residue distances. Does this indicate that the target RMSD moves too quickly during simulation? It would be appreciated if the authors could provide more details about the change that happened to the system during those abrupt changes in residue distances. Instead of following the TMD with a long unbiased MD, would the same mechanism be observed with a gentler TMD where the target RMSD moves more slowly?

I also have the following suggestions regarding the presentation of the figure and table:

1. Table 1 does not specify which systems are in DMG-out conformation.

2. The quality of Fig. 7 needs improvement. Fig. 7 shows four structures, and their positions on the trajectory are labeled in Fig. 7(a). However, this is not explained in the caption. Why are water molecules shown in only three of the four structures? What criteria were used to determine which water molecules to display? The resolutions of residues can be higher to clearly show the interactions between them.

3. Including Fig. 8 earlier in the manuscript would be beneficial as it would provide an overview of the system structure study in this work.

 There are a few typos that need to be corrected:

1. line 104:  date "3/27/2024 2:46:00 PM" seems to be misplaced within the sentence.

2. line 191: it would be Figure 3 instead of Fig 2.

Overall, I found this valuable in providing an atomic-level understanding of the interaction between CDK8 and CycC. I would suggest the publication of this study if the above-mentioned concerns are addressed. 

Author Response

We would like to deeply acknowledge the reviewer for its times and its relevant remarks in order to improve our manuscript. We made below a point-by-point response to all the remarks in the attached pdf file and we hope that they will allow answering the reviewer’s concerns.

Reviewer 2 Report

Comments and Suggestions for Authors

The comments and suggestions are on the attached file.

Comments on the Quality of English Language

Minor editing of English language required.

Author Response

We would like to deeply acknowledge the reviewer for its times and its relevant remarks in order to improve our manuscript. All the modifications required by the reviewer were applied.

Reviewer 3 Report

Comments and Suggestions for Authors

Comments and Suggestions for Authors

 Line 104: “sequence3/27/2024 2:46:00 PM”

Point 2.1 Effect of CycC exclusion on structure and dynamics of CDK8

At the beginning of discussing the results, it would be worth briefly presenting what systems 1a, 1b, 2a, 2b, 3, 4, 6, 7, and 8 are. It is discussed in detail in the Materials and Methods section, but at least transferring the information about the simulated systems from lines 174 -175 to the first paragraph (lines 152-157) would facilitate the data analysis from Table 1. Reference to Figure 2 in line 191 suggests that Figure 2 shows a "loading plot". In line 198, we have a reference to Figure 3B, but in Figure 3, there are no A, B, or C marks for its parts. In Figure S7 (reference on line 212), it is worth pointing out the alphaB-helix, alphaC-helix, and activation loop regions - similar to Figure 1.

You can also find non-English words here (Materiel, exemple, traduce).

Lines 366-380: Please verify that the references to Figure 5 are correct here.

Author Response

(The authors gave the same response as above.)

Reviewer 4 Report

Comments and Suggestions for Authors

16 April 2024

Ms. Ref. No.: ijms-2958960

Journal: International Journal of Molecular Sciences.

Title: Highlighting the major role of cyclin C on Cyclin-dependent kinase 8 activity by molecular dynamics simulations.

Comments:

Thank you for your efforts in composing an on such a pertinent subject. I have taken the liberty of providing you with a few observations that I believe will serve to enhance the quality of your work. Please find my feedback outlined in the following paragraphs

1-      In line 20 of abstract section mentioned “and the structural molecular basis of the protein-protein interaction between the two partners on the other hand” it seems to be better that introduce this line.

2-      In lines 24-27 of abstract section mentioned the role of the he residue Glu99 of the CycC , please add some reasons that were concluded from result section to this.

3-      The results of this article depends on Molecular dynamic simulation (MD simulation), that are conducted under in Silico condition. How can generalized these results?

4-      Moreover,   how can clarify the association between the results of this article to the in vivo condition?

5-      According to section of result and discussion, (Figure 1 and Figure S7)) the activation did not have same some systems in DMG-in conformation, how can clarify it in presence (black) and absence (blue) of the CycC?

6-      According to section of result and discussion, the residue of amino acidsof on Cyclin-dependent kinase 8  between residue 300 to the end din not affected in the presence (black) and absence (blue) of the CycC , how can clarify it?

7-      Which of these regions of Cyclin-dependent kinase 8 (the αC-helix, the αB-helix , the activation loop) has the main role against CycC? And why? 

Author Response

(The authors gave the same response as above.)

Round 2

Reviewer 1 Report

Comments and Suggestions for Authors

I appreciate the authors' effort in addressing my concerns. I am happy to recommend the manuscript for publication.

Reviewer 2 Report

Comments and Suggestions for Authors

Accept in present form.

Reviewer 4 Report

Comments and Suggestions for Authors

nothing

Thanks